# Facile Synthesis of Nano-Flower β-Bi_2_O_3_/TiO_2_ Heterojunction as Photocatalyst for Degradation RhB

**DOI:** 10.3390/molecules28020882

**Published:** 2023-01-16

**Authors:** Mingjun Wang, Che Li, Bingfang Liu, Wenzhen Qin, Yu Xie

**Affiliations:** 1Academy of Art & Design, Nanchang Institute of Technology, Nanchang 330044, China; 2College of Environment and Chemical Engineering, Nanchang Hangkong University, Nanchang 330063, China

**Keywords:** Bi_2_O_3_/TiO_2_, heterojunction, photocatalysts, degradation RhB

## Abstract

Photocatalysis is a hopeful technology to solve various environmental problems, but it is still a technical task to produce large-scale photocatalysts in a simple and sustainable way. Here, nano-flower β-Bi_2_O_3_/TiO_2_ composites were prepared via a facile solvothermal method, and the photocatalytic performances of β-Bi_2_O_3_/TiO_2_ composites with different Bi/Ti molar ratios were studied. The nano-flower Bi_2_O_3_/TiO_2_ composites were studied by SEM, XRD, XPS, BET, and PL. The PL result proved that the construction of staggered heterojunction enhanced the separation efficiency of carriers. The degradation RhB was applied to study the photocatalytic performances of prepared materials. The results showed that the degradation efficiency of RhB increased from 61.2% to 99.6% when the molar ratio of Bi/Ti was 2.1%. It is a mesoporous approach to enhance photocatalytic properties by forming heterojunction in Bi_2_O_3_/TiO_2_ composites, which increases the separation efficiency of the generated carriers and improves photocatalytic properties. The photoactivity of the Bi_2_O_3_/TiO_2_ has no evident changes after the fifth recovery, indicating that the Bi_2_O_3_/TiO_2_ composite has distinguished stability.

## 1. Introduction

Environmental pollution and destruction are becoming a vertical global crisis. Organic dye pollutants are directly discharged into water from dyeing textiles. Most dyes, such as RhB, MB, MR, and EY, are non-biodegradable and carcinogenic, which has brought damage to environment and human health [1]. Therefore, developing an environmental harmony, ecological cleanliness, and safe and energy-saving treatment technology for the problem of environment contamination is the most important immediate challenge that human beings must face [2,3,4]. 

Photocatalysis offers a clean, gentle reaction conditions, an easy procedure, and safe technology that can degrade the pollution under solar irradiation [4,5,6,7]. Photocatalytic technology plays a significant role in the field of environmental pollution protection [8,9,10]. Metal oxide semiconductors have garnered great attention in pollution treatment owing to their cheap and easy synthesis [11,12]. As a classic semiconductor, titanium dioxide (TiO_2_) is the one most generally cited in photocatalytic technology in virtue of its chemical stability, nontoxicity, low price, and high activity [13]. The photocatalytic reaction system and reaction mechanism of TiO_2_ were studied [14,15]. In particular, TiO_2_ photocatalysts have been explored broadly in the area of environmental pollution and energy transformation [16,17,18]. However, the TiO_2_ photocatalysts also expose many problems, such as the large band gap (3.2 eV), absorption narrow wavelength range light, easy recombination with photogenerated electrons, and holes, which bring about lower photocatalytic efficiency, limiting its application [19,20]. For this reason, modification of TiO_2_ for enhancing its photocatalytic properties is needed and crucial. Many different means have been tried by researchers to change the surface or comprehensive performances of TiO_2_, for instance, doping metal [21], structural adjustment [22], and heterogeneous structure [23,24], thereby promoting the photocatalytic properties. In these modifications, construction of heterojunctions is a promising strategy to improve the separation efficiency of photogenerated charge carriers. TiO_2_ united with narrow bandgap semiconductors that are visible-light-responsive would extend the light absorption range and decrease the combination of photogenerated charge carriers, for example, TiO_2_/ZnFe_2_O_4_ [25], CdS/TiO_2_ [26], g-C_3_N_4_/TiO_2_ [27], and InVO_4_/TiO_2_ [28].

Among these visible-light-responsive and narrow band gap semiconductors, bismuth oxide (Bi_2_O_3_) is an up-and-coming semiconductor to construct heterojunctions with TiO_2_, in which the band gap is lower than that of TiO_2_ [29]. When Bi_2_O_3_ is irradiated with visible light, photogenerated holes of the valence band of Bi_2_O_3_ have strong oxidation, which is beneficial to degrade the pollutions. In addition, Bi_2_O_3_ has been under comprehensive study due to its excellent physical and chemical properties, low price, and non-toxicity [30,31,32]. Bi_2_O_3_ has α, β, γ, δ, ε, and ω phases, i.e., six kinds of polymorphs [33]. In particular, β-Bi_2_O_3_ possesses high light-absorption performances among the six polymorphs, and the band gap is about 2.3 eV [34]. Therefore, the construction of β-Bi_2_O_3_/TiO_2_ heterojunction structures can be carried out to increase the light response and photocatalytic performances. However, for the case of β-Bi_2_O_3_/TiO_2_ photocatalytic composites, the preparation methods are complex, including a two-step procedure, photo-deposition, deposition-reduction, and impregnation method [35,36].

Here, in this paper, the β-Bi_2_O_3_/TiO_2_ heterojunction photocatalysts were prepared via a simple, one-step solvothermal method. The influences of the molar ratios of β-Bi_2_O_3_/TiO_2_ on the photocatalytic degradation of RhB for β-Bi_2_O_3_/TiO_2_ were assessed. In the meantime, the morphology, crystal structure, surface chemical optical state, and photoelectrochemical properties of photocatalysts were investigated.

## 2. Results and Discussion

### 2.1. Characterization of Photocatalysts

Figure 1 shows the XRD patterns of pure Bi_2_O_3_, pure TiO_2_, and Bi_2_O_3_/TiO_2_ photocatalysts with different molar ratios of Bi. For the pure TiO_2_, the diffraction peaks are at 25.3°, 37.8°, 48.1°, 53.9°, and 62.7°, coinciding with (101), (004), (200), (105), and (204) of the anatase TiO_2_ (JCPDS No.21-1272) [37]. For pure Bi_2_O_3_, the diffraction peaks are corresponding to β-Bi_2_O_3_ [32]. For Bi_2_O_3_/TiO_2_ composites with different molar ratios of Bi, it is evident that the diffraction peaks are similar to pure TiO_2_ and correspond to the anatase TiO_2_. The diffraction peak of the Bi phase is not observed in Bi_2_O_3_/TiO_2_ composites for several reasons. The size of Bi is extremely small and highly dispersed, and the doping amount of Bi is too small [36]. There are similar results in the reported literatures [38]. For example, Zhu et al. [39] found that Bi_2_O_3_/TiO_2_ lacks the Bi_2_O_3_ diffraction even when the Bi/Ti ratio in Bi_2_O_3_/TiO_2_ was 4.1%.

The morphologies of materials were analyzed via SEM and HRTEM, and the images are displayed in Figure 2 and Figure 3. In Figure 2a, the pure Bi_2_O_3_ reveals a stuffed, coral-like sphere with a diameter between 50–150 nm. The pure TiO_2_ exhibits a flower-like structure (Figure 2b). As shown in Figure 2c–f, the Bi_2_O_3_/TiO_2_ composites in different molar ratios of Bi display a flower-like structure that is similar to the morphology of pure TiO_2_. It indicates that the incorporation of Bi_2_O_3_ does not influence the topography of TiO_2_. In addition, the diameter of the flower-like structure of Bi_2_O_3_/TiO_2_ composites is between 5 μm–10 μm, and the thickness diameter of the flower-like structure is about 65 nm. Meanwhile, the EDS was applied to investigate the elemental contents of 2.1% Bi_2_O_3_/TiO_2_, and the atomic ratio of Bi/Ti is about 2.9%, as displayed in Table 1.

From the TEM images of Bi_2_O_3_/TiO_2_ composites (Figure 3a), some nanoparticles with a diameter between 2~3 nm are evenly spread on the TiO_2_. The HRTEM image of Bi_2_O_3_/TiO_2_ composites (Figure 3b) shows that the lattice fringe is about 0.335 nm, assigned to (110) of β-Bi_2_O_3_. This indicates that β-Bi_2_O_3_ is present in the Bi_2_O_3_/TiO_2_ composites. Moreover, the crystal lattice fringe of 0.365 nm is corresponding to the (101) plane of anatase TiO_2_. This indicates that the Bi_2_O_3_ and TiO_2_ are closely interlinked. Due the deposition of Bi_2_O_3_ on TiO_2_, it could be profitable to the carriers migrate between Bi_2_O_3_ and TiO_2_ and enhance the separation efficiency of carriers as well as the photocatalytic properties. Furthermore, the crystallite size of Bi_2_O_3_/TiO_2_ composites decreased compared with Bi_2_O_3_ and TiO_2_ [40].

The chemical compositions and element valence state of 2.1%Bi_2_O_3_/TiO_2_ photocatalysts were studied through XPS. Figure 4a displays the XPS survey scan of 2.1%Bi_2_O_3_/TiO_2_, which demonstrates that the presence of Bi, O, and Ti elements can be seen in Bi_2_O_3_/TiO_2_ composites. It indicates that the Bi_2_O_3_/TiO_2_ composites managed to compound by solvothermal method. The high-resolution XPS for Bi 4f of Bi_2_O_3_/TiO_2_ composites is displayed in Figure 4b, and the two main peaks at 159.19 eV and 164.43 eV are corresponding to the Bi 4f_7/2_ and Bi 4f_5/2_, respectively, which indicates the presence of Bi^3+^ [41]. In additional, the peaks at 157.43 eV and 162.63 eV are attributed to the Bi 0 (metallic Bi) [42], indicating that the Bi^3+^ and metallic Bi are both in Bi_2_O_3_/TiO_2_ composites. Ti 2p peaks of Bi_2_O_3_/TiO_2_ composites are fitted by two XPS peaks (Figure 4c), and the two peaks at 458.19 eV and 463.81 eV are assigned to Ti 2p_3/2_ and Ti 2p_1/2_, respectively, showing the existence of Ti^4+^ [43]. The high-resolution XPS O1s of Bi_2_O_3_/TiO_2_ composites located at 529.36 eV, 530.04 eV, and 531.71 eV belong to lattice oxygen in Bi-O of Bi_2_O_3_, Ti-O of TiO_2_, and surface absorbed hydroxyl groups, respectively (Figure 4d) [44]. It is worth noting that the binding energies of Ti 2p and O1s in 2.1% Bi_2_O_3_/TiO_2_ are shifted towards lower binding energies compared to TiO_2_, while Bi 4f binding energy is shifted to high binding energy compared to B_2_O_3_. In conclusion, the XPS results indicate strong interactions and electron transfer between Bi_2_O_3_ and TiO_2_ in Bi_2_O_3_/TiO_2_.

The specific surface areas are important elements to influence the catalytic properties. Therefore, the nitrogen adsorption–desorption isotherm was employed to evaluate the BET surface area of β-Bi_2_O_3_, TiO_2_, and Bi_2_O_3_/TiO_2_ photocatalysts, and the results are shown in Figure 5. The isotherms of all photocatalysts are typical type IV, and the H3 hysteresis loop in the light of IUPAC classification can be observed [45]. It indicates that the β-Bi_2_O_3_, TiO_2_, and Bi_2_O_3_/TiO_2_ photocatalysts are mesoporous structures. The BET specific surface areas of β-Bi_2_O_3_, TiO_2_, and Bi_2_O_3_/TiO_2_ photocatalysts are 3.1, 8.7, and 42.0 m^2^/g, respectively. Apparently, the specific surface area of Bi_2_O_3_/TiO_2_ composites is considerably increased after the fusion of Bi_2_O_3_ on TiO_2_ compared with pure TiO_2_. With the increase of calcination temperature, Bi_5_O_7_NO_3_ gradually transformed to β- Bi_2_O_3_, and the following decomposition reaction (Bi_5_O_7_NO_3_ → 5/2Bi_2_O_3_+NO+3/4O_2_) occurs at the calcination temperature. The products of NO and O_2_ during the decomposition of Bi_5_O_7_NO_3_ will affect the crystallization process of TiO_2_, thus making the TiO_2_ structure looser and producing many pores, increasing the specific surface area of the composite. From the SEM results, it can be seen that more holes are formed after the combination of Bi_2_O_3_ and TiO_2_, increasing the specific surface area. In addition, it was reported that the higher specific surface area was frequently accompanied by higher adsorption properties and active sites [46]. Therefore, the Bi_2_O_3_/TiO_2_ composites have a higher adsorption capacity and active sites, indicating the improvement of photocatalytic properties.

### 2.2. Analysis of Optical and Photoelectrochemical Performances

The optical properties of pure Bi_2_O_3_, pure TiO_2_, and Bi_2_O_3_/TiO_2_ composites were assessed through UV–vis DRS. As displayed in Figure 6a, obviously, the absorption edge of TiO_2_ is about 400 nm in the UV spectrum. However, the Bi_2_O_3_ has a larger absorption range, and the absorption edge is about 500 nm. For Bi_2_O_3_/TiO_2_ composites, the absorption edges are red-shifted in comparison to the pure TiO_2_, particularly 2.1% Bi_2_O_3_/TiO_2_ composites. Therefore, after incorporation of Bi_2_O_3_, it is profitable to increase the light energy acquirement and visible light absorption for pure TiO_2_. It would help to separate the photogenerated carriers and increase the properties of degradation of RhB. Further, the Kubelka–Munk formula (αhυ = A(hυ − Eg)^n/2^) was employed to count the band-gap energy of catalysts [47]. The band gap of pure TiO_2_, pure Bi_2_O_3_, and 2.1% Bi_2_O_3_/TiO_2_ samples are about 3.1 eV, 2.72 eV, and 2.79 eV, respectively (Figure 6b).

Photoluminescence (PL) was applied to measure the separating efficiency of photogenerated electrons and holes of prepared samples because of its high sensitivity and because it does not destroy contaminated samples. Figure 7 shows the PL results of pure TiO_2_, 1.3% Bi_2_O_3_/TiO_2_, 2.1% Bi_2_O_3_/TiO_2_, 2.9% Bi_2_O_3_/TiO_2_, 3.7% Bi_2_O_3_/TiO_2_, and 5% Bi_2_O_3_/TiO_2_, respectively. It is evident that the PL intensities of all Bi_2_O_3_/TiO_2_ photocatalysts are less than that of pure TiO_2_, implying that the construction of Bi_2_O_3_/TiO_2_ restrains the regroup of photogenerated electrons and holes. Meanwhile, the 2.1% Bi_2_O_3_/TiO_2_ composite has the lowest intensity, which implies that the separation efficiency of photoinduced carriers is highest, corresponding to the high degradation rate of RhB [45].

### 2.3. Photocatalytic Performance Analysis

The photocatalytic properties of pure TiO_2_, pure Bi_2_O_3_, and Bi_2_O_3_/TiO_2_ composites were assessed by degrading the RhB via simulation sunlight, and the consequences are displayed in Figure 8. The experiment is divided into two steps: first, in order to achieve concentration balance, the photocatalysts adsorbed the RhB for 30 min under dark reactions; then, the photocatalysts were illuminated under simulated sunlight by xenon lamp. Figure 8a presents the impact of Bi_2_O_3_ ratio on TiO_2_ photocatalytic properties. It is found that the pure Bi_2_O_3_, pure TiO_2_, and Bi_2_O_3_/TiO_2_ composites have lower absorption ability in the dark after 30 min. However, the photocatalytic performances of Bi_2_O_3_/TiO_2_ composites with simulated sunlight increased first with the enhancement of Bi_2_O_3_ content, and the 2.1% Bi_2_O_3_/TiO_2_ composites exhibited exceptional photocatalytic abilities. The degradation rate of RhB can reach 99.6% under the irradiation of simulation sunlight for 60 min, which is about 0.63 of the time and six times better than pure TiO_2_ and pure Bi_2_O_3_, respectively. It is mainly because of the constitution of the heterojunction between Bi_2_O_3_ and TiO_2_, which restricts the combination of carriers and thus enhances the photocatalytic properties of the photocatalyst. However, the photocatalytic properties of Bi_2_O_3_/TiO_2_ catalysts declined when increasing the Bi_2_O_3_ content from 2.9% to 5%, which may be the result of the aggregates of the Bi_2_O_3_ and the drop of active sites. On the other hand, the excessive Bi_2_O_3_ can be considered as the recombination center, which results in the decrease of photocatalytic properties. The excessive Bi_2_O_3_ would influence the transmission of light between the Bi_2_O_3_ and TiO_2_, blocking the motivation of TiO_2_ and resulting in reducing the photocatalytic efficiency of photocatalysts [48]. Moreover, the catalytic performances of different Bi_2_O_3_/TiO_2_ reported in other papers were compared, and the details are listed in Table 2. It can be seen that the β-Bi_2_O_3_/TiO_2_ photocatalyst in this work shows the efficient photocatalytic performance for photodegradation RhB.

Meanwhile, the first-order reaction rate constant (k) determined by quasi-first-order kinetic model was applied to study the photocatalytic processes. As shown in Figure 8b, the k values of TiO_2_, Bi_2_O_3_, 1.3%Bi_2_O_3_/TiO_2_, 2.1% Bi_2_O_3_/TiO_2_, 2.9% Bi_2_O_3_/TiO_2_, 3.7% Bi_2_O_3_/TiO_2_, and 5% Bi_2_O_3_/TiO_2_ are 0.03654 h^−1^, 0.0034 h^−1^, 0.04315 h^−1^, 0.07729 h^−1^, 0.05717 h^−1^, 0.05568 h^−1^, and 0.05397 h^−1^, respectively. It is obvious that the k value of 2.1% Bi_2_O_3_/TiO_2_ composites is the maximum, which is 2.1 and 22.7 times of TiO_2_ and Bi_2_O_3_, respectively. The result shows that the 2.1% Bi_2_O_3_/TiO_2_ composites display preferable photocatalytic activity compared to other catalysts. In addition, the recycling stability of the 2.1% Bi_2_O_3_/TiO_2_ composites was assessed. As represented in Figure 8c, the photocatalytic activity of 2.1% Bi_2_O_3_/TiO_2_ composites not to have obvious change after five trials, indicating the outstanding stability and repeatability. Further, in order to verify the stability of 2.1% Bi_2_O_3_/TiO_2_ composites, the XRD was employed to analyze the structure of 2.1% Bi_2_O_3_/TiO_2_ composites before and after five trials, as shown in Figure 8d. It can be seen that there is no obvious change in 2.1% Bi_2_O_3_/TiO_2_ composites before and after five trials, which also implies the stability of Bi_2_O_3_/TiO_2_ composites.

Based on the above analysis, a potential degradation mechanism is put forward. As represented in Figure 9, when the Bi_2_O_3_/TiO_2_ photocatalyst is illuminated by simulated sunlight, the photogenerated electrons (e^−^) are activated and diverted from the CB of TiO_2_ to the CB of Bi_2_O_3_ via the interface of Bi_2_O_3_/TiO_2_ composites; meanwhile, the holes (h^+^) are transferred from the VB of Bi_2_O_3_ to the VB of TiO_2_; therefore, the photoinduced holes accumulate in the heterojunction interface, which is favorable for the separation of photogenerated electrons and holes. The photogenerated electron and holes could have the following reactions [52]: On the one hand, the electrons react with O_2_ in the solution to produce •O_2_^−^. Then, the H+ react with the •O_2_^−^ to generate H_2_O_2_, which then reacts with electrons to transform •OH. On the other hand, the h+ on the VB of TiO_2_ oxides the H_2_O/OH- to form •OH radicals for RhB degradation. Therefore, the Bi_2_O_3_/TiO_2_ composites have higher photocatalytic activity.

## 3. Experimental

### 3.1. Chemicals

Titanium tetrachloride was purchased from Shanghai Nuotai Chemical Co., Ltd, Shanghai, China. Sinopharm Chemical Reagent Co., Ltd, Shanghai, China. provided nitric acid (HNO_3_, AR), bismuth nitrate pentahydrate (Bi(NO_3_)_3_·5H_2_O, AR), and sodium bicarbonate (NaHCO_3_, AR). Sodium hydroxide (NaOH, AR) and glycerol were obtained from Xilong Chemical Co., Ltd, Guangzhou, China. Ethanol (CH_3_CH_2_O, AR) was obtained from Aladdin. Rhodamine B was provided by Shanghai Yuanye Biotechnology Co., Ltd, Shanghai, China.

### 3.2. Preparation of β-Bi_2_O_3_/TiO_2_ Photocatalysts

Bi(NO_3_)_3_·5H_2_O, 30 mL glycerin, and 20 mL ethanol were mixed together and then stirred for 30 min and sonicated for 1 min. The mixtures were poured into a Teflon-sealed reactor. Titanium tetrachloride was added drop by drop into the solution. Then, the Teflon-sealed reactor was put into an oven and heated for 48 h at 110 °C. When the temperature dropped to 25 °C, the reaction productions were collected and washed with ethanol several times and dried at 80 °C for 6 h to obtain the material. The material was calcined at 375 °C for 4 h to obtain the product. The molar ratios of Bi/Ti in the composites were settled at 1.3%, 2.1%, 2.9%, 3.7%, and 5.0%, which were calculated by theoretical methods. The as-prepared photocatalytics were denoted as 1.3%, 2.1%, 2.9%, 3.7%, and 5.0% Bi_2_O_3_/TiO_2_, which are based on the molar ratio of Bi/Ti. As a comparison, pure Bi_2_O_3_ and TiO_2_ samples were prepared with the same procedure.

### 3.3. Characterization

The crystal structure was studied by a X-ray diffraction (XRD, RiGdku, RINT2000 with Cu K α radiation (λ = 0.15418 nm). The morphologies of photocatalysts were studied through a field-induced emission scanning electron microscope (FESEM, JSM-6700F). The transmission electron microscopy (TEM) was employed by JEM-2010F, and the accelerating voltage was 200 kV. The Brunauer–Emmett–Teller (BET) specific surface area of prepared photocatalysts was recorded on a Quantachrome NOVA 2000e. A UV–vis scanning spectrophotometer (UV–vis/DRS, SHIMADZU UV-2450) was applied to investigate the optical properties of the prepared photocatalysts. The photoluminescence (PL) was measured by a Hitachi F4500 fluorescence spectrometer. The X-ray photoelectron spectroscopy (XPS, ESCALAB MK II) was employed to analyze the elemental compositions of the samples.

### 3.4. Photocatalytic Activity Analysis

The photocatalytic properties of all photocatalysts were analyzed by degrading rhodamine B (RhB) with simulation sunlight illumination. The illuminant was a 300 W Xe lamp (PLS-SXE300, Beijing Park Lay Technology Co., Ltd., Beijing, China), and the light intensity was 100 mW/cm^2^. Briefly, the photocatalyst (50 mg) was placed into 60 mL of 20 mg/L of RhB solution. Before exposure to light, the mixture solution was stirred for 30 min to obtain adsorption equilibrium. Then, 3 mL of solution was extracted and centrifuged at 20 min intervals. The RhB concentration was recorded through the UV–vis spectrophotometer. The stability of Bi_2_O_3_/TiO_2_ photocatalyst was studied via five recycling experiments, and the results were the average value of three samples.

## 4. Conclusions

In this study, flower-like Bi_2_O_3_/TiO_2_ photocatalysts were successfully prepared by solvothermal route, and XRD, SEM, TEM, XPS, BET, UV–vis, and PL were employed to analyze the morphology and properties of the photocatalysts. The influence of doped Bi_2_O_3_ content on TiO_2_ photocatalytic efficiency was determined. The XRD results implied that the presence of Bi_2_O_3_ did not destroy the lattice structure of TiO_2_. The photocatalytic properties of materials were studied via RhB degradation. A significantly improvement in photoactivity was obtained when the heterojunction was created between Bi_2_O_3_ and TiO_2_. Further, the 2.1% Bi_2_O_3_/TiO_2_ photocatalyst has the best degradation efficiency, which is 99.6% degradation of RhB at 60 min. It is mainly because the heterojunction in Bi_2_O_3_/TiO_2_ strengthens the movement and separation of carriers and then enhances the photocatalytic properties of the Bi_2_O_3_/TiO_2_.

## Figures and Tables

**Figure 1 molecules-28-00882-f001:**
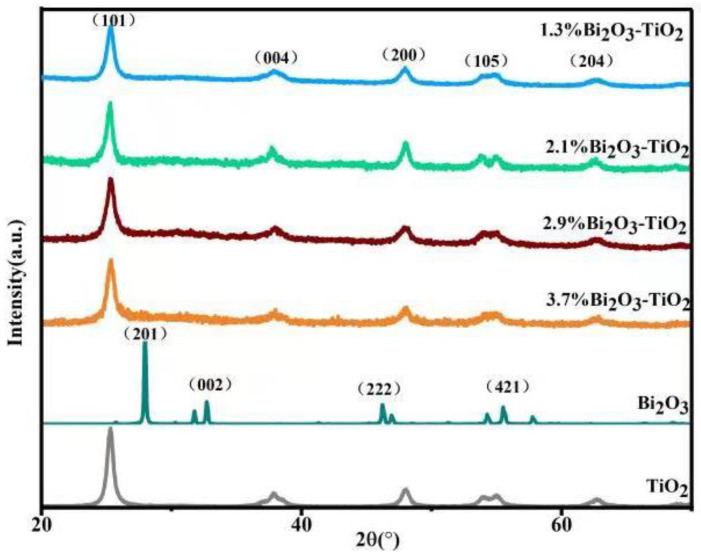
The XRD patterns of β -Bi_2_O_3_, TiO_2_, and Bi_2_O_3_/TiO_2_ composites.

**Figure 2 molecules-28-00882-f002:**
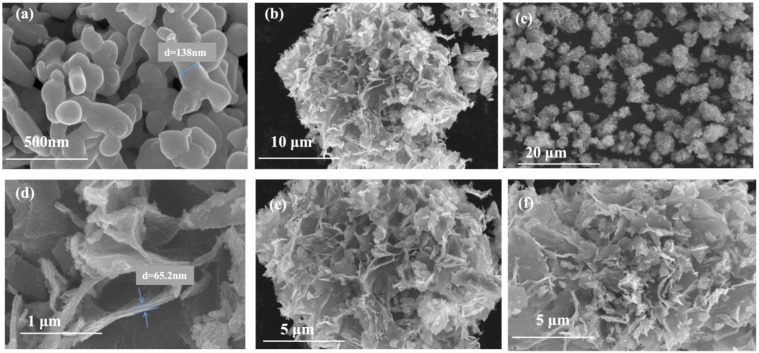
The SEM images of (**a**) β-Bi_2_O_3_, (**b**) TiO_2_, and (**c**) 2.1%Bi_2_O_3_/TiO_2_; (**d**) petal diameter of 2.1%Bi_2_O_3_/TiO_2_, (**e**) 2.9% Bi_2_O_3_/TiO_2_, and (**f**) 3.7% Bi_2_O_3_/TiO_2_.

**Figure 3 molecules-28-00882-f003:**
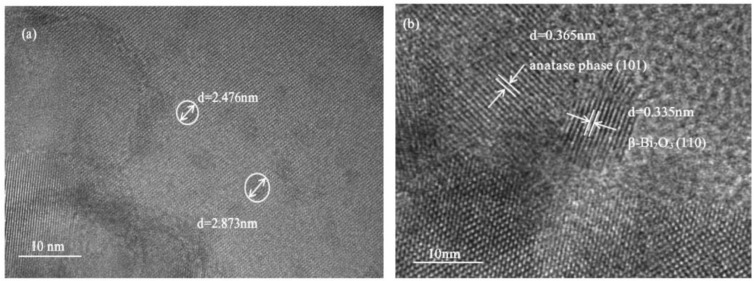
The (**a**) TEM and (**b**) HRTEM of 2.1% Bi_2_O_3_/TiO_2_.

**Figure 4 molecules-28-00882-f004:**
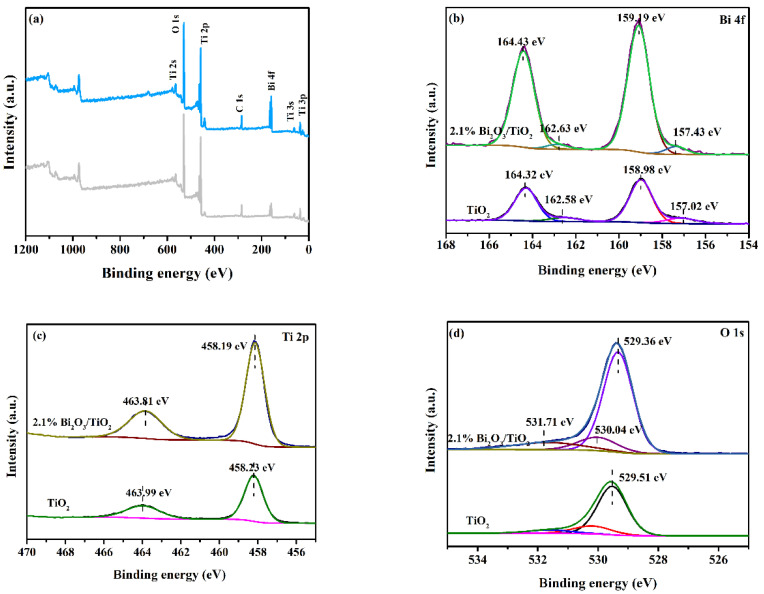
XPS full-spectrum map of 2.1% Bi_2_O_3_/TiO_2_ (**a**) and high-resolution spectrum map of (**b**) Bi 4f, (**c**) Ti 2p, and (**d**) O 1s.

**Figure 5 molecules-28-00882-f005:**
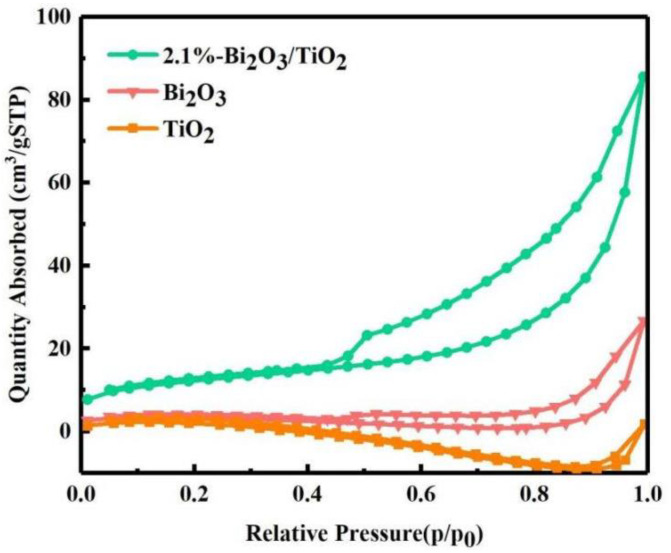
N_2_ adsorption–desorption isotherm of 2.1%Bi_2_O_3_/TiO_2_, Bi_2_O_3_, and TiO_2_.

**Figure 6 molecules-28-00882-f006:**
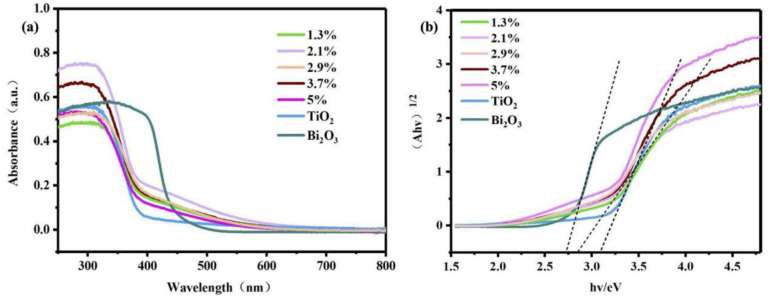
(**a**) UV–vis spectrum and (**b**) the band gap of Bi_2_O_3_/TiO_2_, Bi_2_O_3_, and TiO_2_.

**Figure 7 molecules-28-00882-f007:**
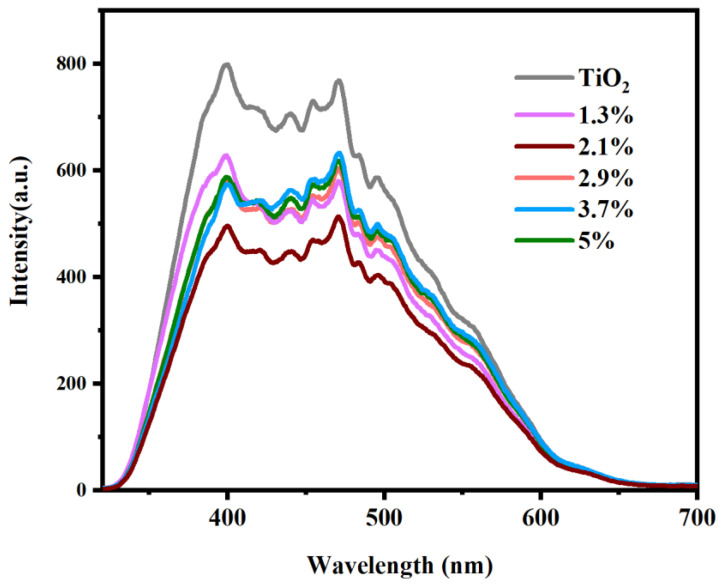
PL fluorescence profiles of TiO_2_ and Bi_2_O_3_/TiO_2_.

**Figure 8 molecules-28-00882-f008:**
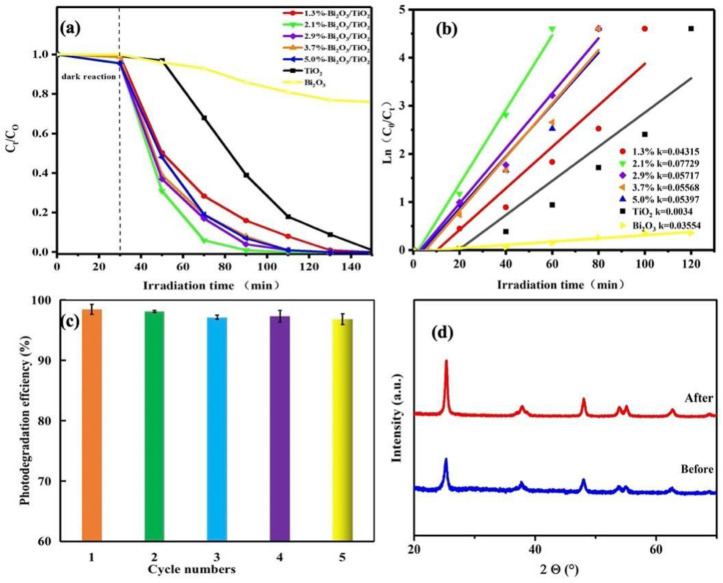
(**a**) Photocatalytic degradation RhB of catalysts, (**b**) reaction kinetic constants for degradation, (**c**) recycle stability of 2.1% Bi_2_O_3_/TiO_2_ composites for RhB degradation, and (**d**) the XRD of before photodegradation and five cycles of photogradation of 2.1% Bi_2_O_3_/TiO_2_ composites.

**Figure 9 molecules-28-00882-f009:**
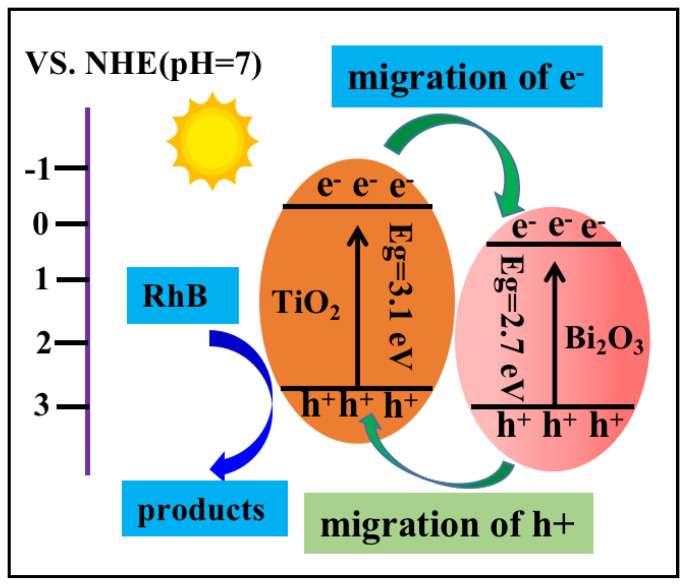
Photocatalytic mechanism based on degradation of the RhB of the Bi_2_O_3_/TiO_2_ photocatalyst.

**Table 1 molecules-28-00882-t001:** The elemental contents of 2.1% Bi_2_O_3_/TiO_2_ from EDS.

Element	Atomic Fraction (%)	Atomic Error (%)	Mass Fraction (%)	Mass Error (%)	Fit Error (%)
O	61.88	1.29	38.71	0.1	0.61
Ti	37.06	0.38	52.26	0.18	2.57
Bi	1.06	9.03	8.03	9.23	0.19

**Table 2 molecules-28-00882-t002:** Comparative performance of Bi_2_O_3_/TiO_2_ materials for photocatalytic dye photodegradation.

Catalyst	Degradation Time (min)	Performance (Efficiency (%))	Light Source	Reference
β-Bi_2_O_3_/TiO_2_	60 min	100%	Simulated sunlight	this work
Bi_2_O_3_/TiO_2_ nanofiber	120 min	65%	Simulated sunlight	[45]
Bi_2_O_3_/TiO_2_-Ph	120 min	87%	Visible light	[35]
Bi_2_O_3_/TiO_2_	75 min	99%	Visible light	[49]
TiO_2_/Bi_2_O_3_-g-C_3_N_4_	120 min	98%	Ultraviolet light/sunlight	[50]
Ag-Bi_2_O_3_-TiO_2_	90 min	100%	Full-spectrum light irradiation	[51]

## Data Availability

Data of the present study are available in the article.

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
