# Peer review of "Facile Synthesis of Nano-Flower β-Bi2O3/TiO2 Heterojunction as Photocatalyst for Degradation RhB"

_molecules, 2023, doi:10.3390/molecules28020882_

Round 1

Reviewer 1 Report

In this work, Xie and Qin synthesized β-Bi2O3/TiO2 heterojunction for photocatalytic degradation of RhB, this work is systematic and I suggested major revision after addressing these comments:

1.     Title, the morphology, nano-flower should be also included in the title;

2.     Line 12, the morphology of nanoflower has been described in Line 10;

3.     Line 13, it’s the staggered heterojunction enhancing the PL efficiency not the existence of TiO2;

4.     Not all the characterizations were mentioned in the abstract and I think the abstract should be revised to emphasize what has been done more logically, not a stack of experimental results;

5.     Keywords, photocatalytic active are all adjective and couldn’t be keywords;

6.     Line 41, enhancing, Line 53, beneficial, Line 64, solvothermal method, Line 77, highly dispersed, et al. the English should be checked;

7.     Line 83, the authors should also emphasize the SEM and TEM could be used to observe the β-Bi2O3 nanoparticles which couldn’t be found by XRD patterns;

8.     Most of the characterizations of 1.3 % and 5.0 % Bi2O3/TiO2 are missing, the authors should explain and added the related data and discussion;

9.     The actual elemental contents should be gave based on the results of XPS;

10.  XPS section, if the synthesized materials could be named as β-Bi2O3/TiO2 heterojunction, rather than β-Bi2O3/Bi/TiO2 heterojunction, the related discussion and mechanism analysis maybe should be revised;

11.  Figure 4c, this graph makes me consider if the author made mistake in subtracting the base line;

12.  Line 181, it could also be ascribed the excessive Bi2O3 acting as the recombination center;

13.  Line 188, the light sources should be listed in the table;

14.  Line 192, the authors mentioned the Bi2O3 was applied the visible light photocatalytic ability of TiO2, why the photocatalysis was still performed under UV regions and pure TiO2 has obvious efficiency?

15.  Line 193, the unit of the kinetic constant is missing;

16.  Figure 8d, the stability of the materials should be verified by the elemental content or XPS spectra of Bi rather than XRD ppatterns;

17.  Figure 9, there is no unit of the y-axis, I don’t know it’s vs. NHE, Fermi level or vacuum level, furthermore, the data sources of the bandgap edges should be clarified, and the radical routes should also be added in this graph.

Reviewer 2 Report

The authors reported the synthesis of Nano-flower β-Bi2O3/TiO2 composites and their photocatalytic performances. They found that the degradation efficiency of RhB increased from 61.2% to 99.6% after the addition Bi2O3 on TiO2. Moreover, the degradation efficiency had a significant improvement when the molar ratio of Bi/Ti was 2.1%. The results are interesting while the organization of data requires further improvement. Here are several questions and suggestions:

l  The abstract part lacks the research motivation. The authors are suggested to add 1-2 sentences about difficulties or problems in this field and explain why their work is valuable.

l  It is strongly suggested to check the grammatical mistakes in the manuscript. For example, For examples, it should be “enhance photocatalytic properties by forming heterojunction” rather than “enhance photocatalytic properties by formation heterojunction” on page 1 line 18.

l  It is well-known that PL is the emission process by recombining electron-hole pairs. The title of the vertical axis in Figure 7 is absorbance. Please explain why?

l  Did authors conduct EDS mapping of as-prepared hetero-structure? How did the authors obtain the molar ratio of Bi/Ti?

Reviewer 3 Report

The manuscript deals with the Facile Synthesis of β-Bi2O3/TiO2 heterojunction as photocatalyst for the degradation RhB. The manuscript is well structured, however, some points should be clear for a better understanding of the presented data for possible publication in Molecules.

A.    In the introduction, the author should clearly mention the utilization of metal oxides and their importance. The author should carefully read the articles like Advanced Powder Technology 32 (2021) 3770–3787; Journal of Nanostructure in Chemistry (2022) 12:547–564 and refine this portion accordingly.

B.     In the introduction section, the author should include the name of targeted dyes, their role, and their hazardous effect on the ecosystem through carefully reading the article Advanced Powder Technology 33 (2022) 103708.

C.     The main focus of authors lies within heterojunction formation, what is the difference between heterojunction and nanocomposite?

D.    The author used SEM/TEM and XRD analysis. Why is it important to determine the crystallite size? I could not find any discussion comparing the results with the crystallite size obtained from XRD analysis with SEM/TEM results. Which technique provides a more accurate value for the average particle size?

E.     The author should find the crystallite size by reading the article Environmental Research 215 (2022) 114140.

F.      Corelate the results of SEM and BET in the manuscript.

G.    The author performed XPS and PL analysis, justify your results of charge transfer from these results.

H.    In mechanism, the author mentioned that the dye was degraded through the production of reactive radicals. There is no justification for the results as EPR and scavenger tests validate the production of reactive radicals. Perform a scavenger test to validate your results.

I.   In photocatalytic test, what was the experimental temperature, and humidity? Why solar light was not used in the experiment? In the introduction, the author wrote about the utilization of solar light.

J.       What is the novelty of the present findings?

Round 2

Reviewer 1 Report

    This work has been improved, and I still have some minor comments:

    1.  Line 15, it's the staggered heterojunction enhancing the separation of charge carriers, not all heterojunctions, such as the included type, the most important point is "staggered", not "heterojunctions".

    2. What's the name of the sample in Response 9? I also suggest the authors add these data in supporting information or add a data availability declaration.

    3. The authors should consider the name of the heterojunctions in the future research.

    4. It's just some words I need to tell the authors, these is potential photodissolution of the doped materials, Bi2O3 might capture electrons and was dissolved. Furthermore, the doped materials cannot be observed by XRD.
